# Stx5-Mediated ER-Golgi Transport in Mammals and Yeast

**DOI:** 10.3390/cells8080780

**Published:** 2019-07-26

**Authors:** Peter TA Linders, Chiel van der Horst, Martin ter Beest, Geert van den Bogaart

**Affiliations:** 1Tumor Immunology Lab, Radboud University Medical Center, Radboud Institute for Molecular Life Sciences, Geert Grooteplein 28, 6525 GA Nijmegen, The Netherlands; 2Department of Molecular Immunology, Groningen Biomolecular Sciences and Biotechnology Institute, University of Groningen, Nijenborgh 7, 9747 AG Groningen, The Netherlands

**Keywords:** syntaxin 5, Golgi apparatus, endoplasmic reticulum, membrane trafficking, secretory pathway

## Abstract

The soluble *N*-ethylmaleimide-sensitive factor attachment protein receptor (SNARE) syntaxin 5 (Stx5) in mammals and its ortholog Sed5p in *Saccharomyces cerevisiae* mediate anterograde and retrograde endoplasmic reticulum (ER)-Golgi trafficking. Stx5 and Sed5p are structurally highly conserved and are both regulated by interactions with other ER-Golgi SNARE proteins, the Sec1/Munc18-like protein Scfd1/Sly1p and the membrane tethering complexes COG, p115, and GM130. Despite these similarities, yeast Sed5p and mammalian Stx5 are differently recruited to COPII-coated vesicles, and Stx5 interacts with the microtubular cytoskeleton, whereas Sed5p does not. In this review, we argue that these different Stx5 interactions contribute to structural differences in ER-Golgi transport between mammalian and yeast cells. Insight into the function of Stx5 is important given its essential role in the secretory pathway of eukaryotic cells and its involvement in infections and neurodegenerative diseases.

## 1. Introduction

The secretory pathway is essential for secretion of cytokines, hormones, growth factors, and extracellular matrix proteins, as well as for the delivery of receptors and transporters to the cell membrane and lytic proteins to endo-lysosomal compartments. Proteins destined for the secretory pathway are synthesized at the endoplasmic reticulum (ER) and subsequently transported to their destination by vesicular trafficking via the *cis*- to medial- to *trans*-Golgi cisternae and finally to the *trans*-Golgi network [1,2,3]. ER-Golgi transport has been mostly studied in mammalian cells and the yeast *Saccharomyces cerevisiae*, and although the basic mechanisms of this ER-Golgi trafficking are well conserved among eukaryotic cells, there are three pronounced differences between the yeast *S. cerevisiae* and mammalian cells (Figure 1). The first difference concerns the spatial organization of the Golgi apparatus. In most mammalian cell types, a single large Golgi apparatus is juxtaposed with the nucleus and surrounds the microtubule organizing center (MTOC) [2,4,5]. In contrast, in *S. cerevisiae*, the Golgi is organized into discrete cisternae, consisting of individual cisternae that are scattered throughout the cytoplasm, while in other yeast, such as budding *Pichia pastoris* and *Schizosaccharomyces pombe*, the Golgi is present as mini-stacks that are dispersed in the cytoplasm [4,5]. The second difference is the presence of an intermediate compartment between the ER and *cis*-Golgi in mammals called the ER-Golgi intermediate compartment (ERGIC) or vesicular-tubular cluster (VTC) [6]. Yeast does not have an ERGIC, and anterograde trafficking from the ER occurs directly to the *cis*-Golgi [7] via vesicles coated with the cage protein complex COPII, while retrograde trafficking in the reverse direction occurs via vesicles coated with COPI [2,4]. In mammalian cells, anterograde trafficking from the ER also occurs via COPII-coated vesicles, but in this case, proceeds to the ERGIC [2,4,5]. In mammalian cells, not only retrograde trafficking from the ERGIC back to the ER but also further anterograde trafficking to the *cis*-Golgi might occur via COPI-coated vesicles [8]. A final difference in ER-Golgi trafficking is the involvement of microtubules in mammalian cells, but not in yeast [2,4,5]. In this review, we argue that these mechanistic differences between mammals and yeast are partly attributable to one of the central players in ER-Golgi trafficking: the SNARE protein syntaxin 5 (Stx5) in mammals and its yeast ortholog Sed5p.

## 2. SNARE Proteins in ER-Golgi and Intra-Golgi Transport

The SNARE protein family consists of about 38 members in humans and about 24 in yeast and is responsible for most intracellular fusion events of organellar trafficking [9,10,11]. The central hallmark of SNARE proteins is the presence of one or two SNARE motifs of 50–70 residues in size. Based on the structures of these motifs, SNAREs can be grouped into R-SNAREs, with an arginine residue located at the center of the SNARE-motif, and Qa-, Qb- and Qc-SNAREs, with a central glutamine residue [10,11]. Membrane fusion requires three or four cognate SNARE proteins that together contribute four SNARE motifs, one of each group (R, Qa, Qb, Qc). For membrane fusion, these cognate SNARE proteins need to be anchored to both the donor membrane (e.g., COPII vesicle) and acceptor membrane (e.g., *cis*-Golgi) via a C-terminal transmembrane helix or by lipid modifications [10,11]. SNAREs in the donor membrane are called v-SNAREs (vesicular-SNAREs) and SNAREs in the acceptor membrane t-SNAREs (target-SNAREs). Cognate SNARE proteins can form a tight α-helical coiled-coil bundle, called the SNARE complex, which overcomes the energy barrier of membrane fusion. In addition to this, the fusogenic activity of ER-Golgi SNAREs is tightly regulated by numerous interacting proteins and by N-terminal-regulating motifs, which are present on most SNAREs [10,11]. After membrane fusion, the SNARE complex is disassembled by the AAA ATPase *N*-ethylmaleimide-sensitive factor (NSF) in mammals and Sec18p in yeast, which are recruited by the adaptor protein-soluble NSF-attachment protein α (α-SNAP) in mammals and Sec17p in yeast [10,11].

In mammalian cells, the Qa-SNARE Stx5 is an integral component of ER-derived COPII transport vesicles and is required for the docking and fusion of these vesicles to assemble the ERGIC [6,12] by forming a SNARE complex with GosR2 (GS27, membrin; Qb-SNARE) or GosR1 (GS28; Qb), Bet1 (Qc), and Ykt6 (R) or Sec22b (Ers24; R) (Figure 2) [13,14,15,16,17,18,19]. This process is well conserved in yeast, where the fusion of COPII vesicles is believed to occur directly at the Golgi [7] by interactions of the orthologs of the mammalian SNAREs Sed5p (Qa), Bos1p (Qb), Bet1p (Qc), and Sec22p (R) [20,21,22,23]. Stx5/Sed5p is not involved in COPI-mediated retrograde transport from the Golgi to the ER, but this is mediated by Stx18 (Qa), Sec20 (Qb), Use1 (Qc), and Sec22b (R) in mammals and Ufe1p (Qa), Sec20p (Qb), Use1p (Qc), and Sec22p (R) in yeast [3,24,25]. However, Stx5 is involved in COPI-dependent intra-Golgi retrograde trafficking between cisternae and/or retrograde trafficking from endosomes to the *trans*-Golgi network by forming a SNARE complex with GosR1 (Qb), Bet1L (GS15; Qc), and Ykt6 (R) [3,14,17,26,27,28]. In yeast, intra-Golgi retrograde trafficking is mediated by Sed5p forming a SNARE complex with Gos1p (Qb), Sft1p (Qc), and Ykt6p (R) [3,23,29]. Anterograde vesicular trafficking is in principle not required for intra-Golgi transport, as in the widely-accepted cisternal maturation model, the cisternae are very dynamic entities, constantly forming at the *cis*-side by homotypic or/and heterotypic vesicular fusion, and disassembling into anterograde and retrograde membrane carriers at the *trans*-side [2]. In this model, all intra-Golgi vesicular traffic is going in a retrograde direction to recycle Golgi enzymes, cargo receptors and SNAREs [2]. In contrast, according to the vesicular transport model, the cisternae are relatively static with a constant enzyme composition. In this case, the cisternae accept anterograde vesicles with cargo molecules and shed retrograde vesicles with recycling SNAREs and cargo receptors [2]. In favor of the vesicular transport model, Stx5 was found to mediate anterograde trafficking within the Golgi, at least in *Drosophila melanogaster* [30].

Both knockdown and overexpression of Stx5 induce Golgi fragmentation in mammalian cells [15,31,32] and *D. melanogaster* [33]. Because ER-Golgi transport is at the base of most exocytic trafficking, knockdown of Stx5 also causes downstream defects in lysosomal trafficking and autophagy [34]. In yeast, Sed5p is rate-limiting for ER-Golgi transport, because overexpression of Sed5p and to a lesser extent other SNAREs involved in ER-Golgi trafficking (Bos1p, Bet1p, Sec22p) resulted in higher secretion of overexpressed cellulase [35]. Most importantly, Stx5/Sed5p is an essential protein, and Stx5/Sed5p knockout is lethal for mammalian cells [36] and yeast [37], while the STX5 gene knockout is not viable for mice [38] and *D. melanogaster* [39].

There are two notable differences between the function of SNAREs in ER-Golgi transport and exocytic and endocytic trafficking. First, in mammals, Bet1 is proposed to be the v-SNARE for anterograde trafficking [3,19,26,27,28], although another study reported that this is Sec22b [13], and Bet1L is the v-SNARE for retrograde trafficking in intra-Golgi transport [3,19,26,27,28]. Similarly, in yeast, the v-SNAREs for antero- and retrograde traffic are the yeast orthologs Bet1p and Sft1p, respectively [23,29,40]. This differs from other described SNARE-mediated trafficking routes, where R-SNAREs generally act as v-SNAREs, e.g., synaptobrevin in exocytosis, and the Qa-, Qb-, and Qc-SNAREs form an acceptor t-SNARE complex in the target membrane [9,10,11]. The second difference is in the promiscuity of SNAREs. Most SNAREs involved in exocytic and endocytic trafficking are very promiscuous, and nearly every combination of a Qa-, Qb-, Qc-, and R-SNARE can form a SNARE complex in vitro [9]. Because of this promiscuity, exocytic and endocytic SNAREs are often functionally redundant, and knockout or knockdown mostly has no or a very minor phenotype [9]. This is different for Stx5/Sed5p and other ER-Golgi SNAREs, which have been shown in in vitro studies to be highly stringent with purified Sed5p only being able to form a SNARE complex with other SNAREs involved in Golgi-ER trafficking Gos1p (Qb), Bos1p (Qb), Sft1p (Qc), Bet1p (Qc), Ykt6p (R), and Sec22p (R) [41,42]. In line with this, Sed5p forms two distinct non-overlapping Golgi SNARE complexes in vivo (Sed5p-Bos1p-Bet1p-Sec22p and Sed5p-Gos1p-Ykt6p-Sft1p) [23].

In contrast to Stx5, which distributes evenly throughout the Golgi stack, electron microscopy revealed that Bet1L and Bet1 have opposite distributions within the Golgi stack with Bet1L more present at the *trans*-Golgi and Bet1 more present at the *cis*-Golgi [27]. This finding is the base of the SNARE gradient model, where retrograde trafficking is mediated by SNARE complex formation of Bet1L with Stx5, GosR1, and Ykt6 and anterograde trafficking by complex formation of Bet1 with Stx5, GosR2, and Sec22b [1,3,27]. Electron microscopy also revealed that within each Golgi layer, the SNAREs are also heterologously distributed with Bet1L, Stx5, and GosR1 more located to the center of each stack, whereas Bet1, Sec22b, and GosR2 locate more to the rims where they can be more efficiently incorporated into transport vesicles [43]. As Bet1 and Bet1L mediate anterograde and retrograde intra-Golgi trafficking, respectively [3,19,26,27,28], this finding suggests that SNAREs involved in retrograde trafficking might be Golgi resident, whereas the SNAREs involved in anterograde trafficking might dynamically cycle through the Golgi layers. However, it is unclear how this relates to the cisternal maturation model that mainly relies on retrograde vesicular transport and in principle does not require any anterograde intra-Golgi vesicular transport [2].

Stx5 is also involved in ER-Golgi trafficking of specialized cargo molecules. First, Stx5 is required for ER exit of pro-collagens, which are too large to fit in COPII transport vesicles, because its knockdown results in a reduction of pro-collagen I and VII secretion and accumulation of these pro-collagens in the ER [44]. Second, Stx5 mediates the specific export of very low-density lipoproteins (VLDLs) from the ER. VLDLs are synthesized in the ER of liver cells and then transported to the Golgi prior to secretion at the plasma membrane. This transport occurs via unique ER-derived vesicles called VTVs (VLDL-transport vesicles) [45]. These vesicles differ from the other protein exporting vesicles as they are larger, have a lower buoyant density, and have different cargo and protein compositions [45]. It was found that VTVs contain Sec22b, which is able to make a SNARE complex at the *cis*-Golgi together with Stx5, GosR1, and Bet1 [46]. Stx5 is able to bind to the cytoplasmic C-terminal domain of the VLDL receptor (VLDL-R), thereby influencing the receptor’s glycosylation and trafficking [47]. Overexpression of Stx5 was found to prevent Golgi-maturation of VLDL-R and resulted in the decreased presence of VLDL-R at the *trans*-Golgi [47]. However, VLDL-R did not accumulate at the ER nor Golgi and was still translocated from the ER to the cell surface, indicating that overexpression of Stx5 resulted in circumvention of the regular secretory pathway and immature VLDL-R reached the cell surface via an alternative pathway [47]. In addition to VLDL, Stx5 mediates trafficking of another lipoprotein in intestinal cells. Dietary long-chain fatty acids are esterified to triacylglycerol and packaged in the chylomicron, which is the unique lipoprotein of the intestine. Rate-limiting for the transit of chylomicrons through the enterocyte is the exit of chylomicrons from the ER to the *cis*-Golgi in specialized 250 nm-sized transport vesicles. This process involves a t-SNARE complex of Stx5, Vti1a, and Bet1 and the v-SNARE VAMP7, which, based on inhibition of these SNAREs with antibodies, is believed to mediate the fusion of the chylomicrons with the *cis*-Golgi [48]. Note that in contrast to canonical ER-Golgi transport, where Bet1 acts as the v-SNARE [3,19,26,27,28], in these lipoprotein trafficking routes, Bet1 is believed to act as a t-SNARE together with Stx5. However, this has never been studied side-by-side with the same assays, and whether the dual function as Bet1 as both a t- and v-SNARE is true remains to be established.

## 3. Subcellular Localization of Stx5 Isoforms

In mammalian cells, translation of Stx5 can occur at two different starting methionines resulting in two distinct isoforms: a 34.1-kDa short isoform of 301 residues and a 39.6-kDa long isoform that is extended by 54 N-terminal residues (Figure 3) [49]. In yeast, only a single 38.8-kDa and 340 residue-long isoform of Sed5p is expressed, which corresponds to the short isoform of Stx5 in mammalian cells. Both the short and long Stx5 isoforms in mammalian cells and yeast Sed5p are so-called tail-anchored proteins that contain a C-terminal transmembrane helix, which after translation is inserted into the ER membrane by Asna1 in mammals [50] and its yeast ortholog Get3p [51] of the tail recognition complex (TRC) pathway. The subcellular localization of Stx5 and Sed5p at the ER-Golgi interface is largely determined by its transmembrane helix. Stx5 and Sed5p have a short transmembrane helix of only 20 residues in length, which is much shorter than most other SNAREs (~24 residues), and this supports their recycling to the *cis*-Golgi and ER [52,53]. The membranes of the ER and Golgi have thinner membranes than other organelles, because they contain less cholesterol, and thereby provide a better matching with the short lengths of the short transmembrane helices of Stx5 and Sed5p [53]. In fact, overexpression of truncation constructs revealed that the transmembrane domains of Stx5 and Sed5p are sufficient for proper localization to the *cis*-Golgi [54,55,56]. Both isoforms of mammalian Stx5 and yeast Sed5p contain an N-terminal Habc domain, which, at least for Stx5, can interact with its SNARE motif and thereby inhibit SNARE complex formation [13], similar to many other syntaxins [9].

Although both the short and the long isoforms of mammalian Stx5 can be found at ER-derived COPII vesicles [6], the long isoform caries a double-arginine ER retrieval motif, which makes it localize more at the ER (Figure 3) [49,57,58]. The short isoform of mammalian Stx5 lacks this ER retrieval motif and can be found more at the Golgi [49,57,58]. Yeast Sed5p does not contain an ER retrieval motif, and when it is heterologously expressed in mammalian cells Sed5p, locates to the *cis*-Golgi and ERGIC, similar to the short isoform of Stx5 [55]. In yeast, Sed5p localizes to COPII vesicles, and Sed5p immune-isolated vesicles carry early Golgi mannosyltransferases (Mnt1p, Van1p, and Mnn9p), whereas late Golgi and ER proteins are almost completely absent [59].

The presence of an ER retrieval motif argues that the long isoform of Stx5 might be more involved in retrograde transport, while the short isoform mediates anterograde transport. However, this might be too simplistic, as immunoprecipitation revealed that Bet1L, the v-SNARE of retrograde intra-Golgi trafficking, forms a SNARE complex mainly with the short and less with the long isoform of Stx5, whereas GosR1, which has both retro- and anterograde trafficking roles, exclusively interacted with the short Stx5 isoform [16,67], indicating that retrograde Golgi transport is mainly mediated by the short isoform of Stx5. Instead, the long form of Stx5 is involved in the regulation of the ER structure by linking the ER membrane to microtubules (Figure 3) [68] possibly via the adapter protein CLIMP-63 [68,69]. In addition to regulation of ER structure, Stx5 has been implicated in calcium storage at the ER. Both the long and short isoform of Stx5 directly interact with the calcium channel polycystin-2 (PC2) via their SNARE motifs, and this blocks channel activity and prevents calcium leakage from the ER [63].

The localization of both the long and short isoform of mammalian Stx5 and yeast Sed5p is regulated not only by the presence or absence of an ER retrieval motif and their transmembrane helices but also by a number of protein-protein interactions. First, Stx5 is regulated by other SNAREs, because in mammalian cells, overexpression of mutant forms of Bet1L, the v-SNARE of retrograde trafficking, results in altered distribution of Stx5, its SNARE partner GosR1, and the medial-Golgi protein Golgi mannosidase II to *cis*-Golgi and ER [26], possibly because of increased retrograde transport of Stx5. Second, the subcellular localization of Stx5 is mediated by the COG (conserved oligomeric Golgi) tethering complex as experiments targeting the Cog4 subunit of COG to mitochondria showed that this interaction resulted in delocalization of Stx5 to mitochondria [70]. Third, in both mammals and yeast, the sorting of Stx5 and Sed5p to COPII vesicles is mediated by the COPII subunit Sec24. In mammals, Sec24 is present in four different isoforms from gene duplication, called Sec24A–D. Sec24A and B recruit Sec22b, whereas Sec24C and D recruit a Q-SNARE complex of Stx5, GosR2, and Bet1 to COPII-coated vesicles [20,62], but the functional role of this differential binding is still unclear. In yeast, only a single form of Sec24p is present that interacts with Sed5p via two distinct sites (Figure 3 and Figure 4A) [71,72]. The first site of Sec24p, called the A-site, binds to a YNNSNPF motif of Sed5p (residues 203–209) [72] and is not conserved in mammalian Sec24 isoforms (Figure 3 and Figure 4A). Binding of Sec24p to a pre-assembled t-SNARE complex of Sed5p, Bos1p, and Sec22p is favored compared to single Sed5p, as this assembly results in a conformation change that exposes the YNNSNPF motif [72]. The second site of Sec24p, called the B-site, binds to a LxxME motif present in both Bet1p and Sed5p (residues 238–242) [72]. The LxxME motif seems conserved in Stx5 (Figure 3), and the B-site seems conserved in Sec24A and B (not in C and D) (Figure 4A), suggesting that mammalian Sec24A and B can also bind to Stx5, but this has not been proven. The recruitment of mammalian Stx5 to Sec24C and D occurs by direct interactions of the open form of Stx5 (thus the N-terminal Habc-domain not bound to its SNARE motif) to the IxM cargo-binding site of Sec24C or D via a conserved region on Stx5 (residues 242–249; Figure 3 and Figure 4B) [20,62]. This recruitment of Stx5 to COPII-coated vesicles, therefore, differs markedly from the recruitment of Sed5p in yeast.

## 4. Posttranslational Modifications

Stx5 is dynamically regulated during the cell cycle by ubiquitination. During mitosis, particularly the short isoform of Stx5 is monoubiquitinated at K270 (K324 in the long isoform) [65] by HACE1, and this prevents its complex formation with Bet1 and results in Golgi fragmentation [65]. After mitosis, Stx5 is deubiquitinated by VCIP135, enabling SNARE complex formation with Bet1 and reassembly of the Golgi fragments [65,73]. K324 is conserved in yeast Sed5p (K310) (Figure 3), but since the nucleus remains intact during budding of yeast, it is a question if Sed5p is regulated in a similar manner as Stx5 in mammals. In mammalian cells, Golgi reassembly after cell division is also regulated by interactions of Stx5 with the triple AAA ATPase p97/VCP (valosin-containing protein) subunit p47 [73,74]. The activity of Stx5 is not only blocked during mitosis but also in cells undergoing apoptosis. In this case, Stx5 is cleaved by caspase-3 at conserved D263 resulting in a 26-kDa product (Figure 3) [64], which might well be the same as the prominent 31-kDa breakdown product originally reported [16]. Finally, in yeast, Sed5p is regulated by phosphorylation at serine 317, presumably by protein kinase A [66], which is conserved in mammalian cells (human: S331; Figure 3). The phosphomimetic mutant (serine to aspartate) of Sed5p results in enlarged ER, disruption of Golgi trafficking, and impaired cell growth, whereas the phospho-dead mutant (serine to alanine) does not affect ER-Golgi transport nor cell growth, but results in enlargement of the Golgi [66]. The precise role of this phosphorylation of Sed5p is not known, but it is speculated that it might play a role in Golgi inheritance during mitosis, as it allows for the Golgi to cycle between ordered and dispersed states [66].

## 5. Scfd1/Sly1p

ER-Golgi transport and trafficking within the Golgi stack are regulated by the Sec1/Munc18-like protein Scfd1 in mammals and its ortholog Sly1p in yeast. Sec1/Munc18-like proteins regulate the assembly and activity of SNARE complexes in membrane fusion events [75], and in mammals, Scfd1 functions with Stx5 in ER-Golgi trafficking and also functions in the assembly of pre-Golgi intermediates through interactions with Stx17 and Stx18 [6,60,61,76,77,78]. In yeast, Sly1p is implicated in antero- and retrograde ER-Golgi trafficking [77]. Scfd1/Sly1p binds to Stx5/Sed5p in the Golgi via a well-conserved N-terminal region upstream of the Habc-domain (Figure 3) [60,61,76,77]. This N-terminal region is also present in mammalian Stx17 and Stx18 and in yeast Ufe1p (Stx18 ortholog), which also bind Scfd1/Sly1p. In mammalian cells, Scfd1 regulates ER-Golgi anterograde transport via the assembly of pre-Golgi Stx5 SNARE intermediates [6,76,78] and retrograde trafficking via association with the Cog4 subunit of the COG tethering complex [79]. Scfd1 is also involved in a secretory pathway that is independent of COPII, as in zebrafish chondroblasts, the loss of Scfd1 blocks the transport of type II collagen from the ER [80]. Scfd1 mediates pro-collagen export from the ER in mammalian cells as well by binding to the protein TANGO1 [44].

The crystal structure of yeast Sly1p bound to the N-terminal fragment of Sed5p has been resolved [61] and an NMR structure for mammalian Scfd1 bound to the short isoform of Stx5 [60], showing that the high-affinity binding of the N-terminal fragment of Stx5/Sed5p to Scfd1/Sly1p is well conserved (Figure 3 and Figure 4C). Given the conservation and location of this interaction site, Scfd1 is predicted to bind to both the short and long isoforms of Stx5, and Scfd1/Sly1p is predicted to bind to both unbound Stx5/Sed5p and to Stx5/Sed5p in complex with other SNAREs [2,81]. However, at least for yeast Sly1p, this high-affinity binding to Sed5p is not required for its regulation of ER-Golgi transport, but this is mediated by a second lower affinity binding to Sed5p and other SNAREs [82]. This second binding mode might accelerate the assembly of the Sed5p-Bos1p-Bet1p-Sec22p SNARE complex by releasing the Habc-domain from the SNARE motif of Sed5p [83,84], although in another study, they found no significant difference in the kinetics of SNARE complex formation [81]. In mammalian cells, Scfd1 might have no effect on SNARE complex formation as well, because although it is required for ER-Golgi transport, Scfd1 does not promote accessibility of the Stx5 SNARE motif for an antibody that only binds Stx5 in its open conformation [85]. An alternative, or complementary, explanation for how Sly1p regulates ER-Golgi transport is by preventing the interactions of Sed5p with non-cognate SNAREs, which would avert the formation of non-productive SNARE complexes [81]. In line with this, overexpression of Scfd1 in mammalian cells was found to neutralize the dominant-negative effects of excess Stx5 on ER-Golgi trafficking [78]. Finally, Sly1p/Scfd1 might prevent the dissociation of partly-formed SNARE complexes prior to fusion, as in yeast, Sly1p co-assembles with the α-SNAP ortholog Sec17p to prevent disassembly of the Sed5p-Bet1p-Bos1p-Sec22p SNARE complex by the NSF ortholog Sec18p [86]. In contrast to Ufe1p, another interacting SNARE of Sly1p, Sly1p binding does not affect the stability of Sed5p [87]. Both the short and long isoform of Stx5 also interact with the chaperone hsc70, and at least in vitro, this promotes SNARE complex formation of Stx5, GosR2, Bet1, and Sec22b [88].

## 6. Tethering Complexes

Prior to SNARE complex formation, Golgi transport vesicles are captured to the target organelle by two types of membrane tethering complexes [89,90]. The first family consists of long coiled-coil proteins and includes the Golgins p115 and GM130, which mediate ER-Golgi transport, intra-Golgi transport, and Golgi biogenesis [2,90]. The second family consists of multi-subunit tethering complexes of a heterogeneous structure and composition and includes the COG complex, which organizes vesicle tethering in intra-Golgi retrograde trafficking [2,89]. These tethering complexes regulate the specificity of Golgi trafficking by interacting with SNARE proteins, Rab-GTPases, and COPI and COPII vesicle coats, and this, in turn, facilitates the complex formation of cognate SNARE proteins [2,3,89,90].

In mammalian cells, the COG complex has been shown to interact with Stx5 and other Golgi SNAREs, and this interaction enhances the fusogenic assembly of SNARE complexes [67,91,92,93]. The COG complex is composed of eight different subunits called Cog1–Cog8. Cog1–Cog4 are organized in lobe A, which is mainly located at the Golgi stacks, whereas Cog5–Cog8 form lobe B, which locates predominantly at vesicle-like structures [94]. The COG complex is well conserved in yeast [2], and yeast COG interacts with Sed5p [67,95]. In mammalian cells, Stx5 interacts with Cog6 [67,70,93] and Cog8 [70], and these interactions are required for the assembly of the Stx5-GosR1-Bet1L-Ykt6 SNARE complex [92]. However, the best-characterized COG interactions of Stx5 are with subunit Cog4 [67,70,79,93]. Cog4 interacts with the SNARE domain of the short isoform of Stx5 [67], and as this domain is present in both Stx5 isoforms, it seems likely that Cog4 interacts with the long isoform of Stx5 as well. Adjacent to the Stx5 binding site of Cog4 is a site that interacts with Scfd1, and this promotes SNARE complex formation [67]. Knockdown of COG subunits or overexpression of a dominant-negative Cog4 fragment containing the Stx5 and Scfd1 binding sites results in blockage of Golgi-ER retrograde trafficking, mislocalization of Stx5, GosR1, and Bet1L away from the Golgi, and impaired SNARE complex formation of the short Stx5 isoform with GosR1, Bet1L, and Ykt6, whereas SNARE complex formation of the short and long isoforms of Stx5 with Bet1 is not affected [67,79,93].

The Golgins p115 and GM130 also interact with Stx5. The C-terminal region of p115 contains four coiled-coil domains (CC1–CC4), and the first coiled-coil domain CC1 bears weak sequence homology to a SNARE motif and can bind to Stx5 [96]. In addition to Stx5, CC1 can bind to other SNAREs involved in both antero- and retrograde Golgi trafficking (GosR1, GosR2, Ykt6, Sec22, Bet1, Bet1L) and to Scfd1 [90]. These interactions of p115 with Stx5 and other ER-Golgi SNAREs enhance in vitro SNARE complex assembly [96]. p115 is conserved in yeast, and its ortholog Uso1p binds to Sed5p by a region containing CC1 and CC2 [97]. Not only CC1, but also CC4 are required for Golgi trafficking by interacting with multiple Golgi SNAREs (GosR1, GosR2, Ykt6, Bet1, and Bet1L) [90]. Thereby, p115 might connect t- and v-SNAREs in opposing membranes [90] and facilitate SNARE complex formation preceding membrane fusion [1]. Knockdown of p115 results in Golgi fragmentation and blocks ER-Golgi trafficking [1]. p115 also interacts with the Golgin GM130, which in turn also binds to Stx5 [98] via a membrane-proximal region of GM130 [99]. However, in this case, GM130 binding to Stx5 is proposed to prevent its interaction with other SNAREs [99]. Binding of p115 to GM130 releases Stx5 from GM130 and allows SNARE complex formation and membrane fusion downstream of tethering [99]. During mitosis, the interactions of GM130 with p115 are inhibited, whereas binding of GM130 to Stx5 is increased, likely due to phosphorylation of GM130, which would reduce membrane fusion, and hence suggests a role for GM130–Stx5 interactions in the disassembly of the Golgi during cell division [99].

## 7. Infections and Neurodegenerative Disease

Stx5 is involved in neurodegenerative diseases and infections of intracellular parasites and viruses. This makes it a therapeutic target, and in a screen for small molecule inhibitors of retrograde Golgi-ER transport of Shiga toxin, the compounds Retro-1 and Retro-2 were identified, which result in mislocalization of Stx5 away from the Golgi apparatus [100,101]. Stx5 localizes to vacuoles containing the intracellular pathogen *Leishmania amazonensis*, where it mediates the communication with early secretory vesicles, and knockdown or blockage of Stx5 using the small molecule inhibitor Retro-2 was found to reduce the size of these parasitophorous vacuoles and impair *Leishmania* replication and infection [100,102]. Stx5 has also been described to play a role in viral infections. First, infection with human cytomegalovirus (HCMV) results in increased cellular levels of Stx5, which is recruited to the viral assembly site and is required for the efficient production of viral particles [103]. Second, Stx5 is required for infection with adeno-associated virus (AAV), although here, it is likely responsible for viral transport from endosomes to the *trans*-Golgi network [104].

Both in Alzheimer’s and Parkinson’s disease, Stx5 has been implicated. The short and the long isoforms of Stx5 mediate ER-Golgi transport of presenilin 1 and 2, which are subunits of the γ-secretase that cleaves β-amyloid pre-protein (APP) and is responsible for β-amyloid peptide production [57,105,106]. Stx5 interacts directly with presenilin 1 and 2 via both its Habc regulatory domain and transmembrane helix, and these interactions are reduced in a mutant of presenilin 1 that is associated with Alzheimer’s disease [57,105,106]. Moreover, Stx5 expression is upregulated in neurons under ER and Golgi stress [107], and overexpression of the short, but not long, isoform of Stx5 causes accumulation of APP in the ER and inhibits the formation of β-amyloid peptides [57,106,107]. Stx5 may also play a role in Parkinson’s disease, as a disease-related mutant of α-synuclein (A53T) binds to Stx5 and GosR2, and this reduces the formation of the Stx5-GosR2-Bet1-Sec22b SNARE complex, thereby possibly impairing ER-Golgi transport [108].

## 8. Discussion

Despite the high conservation in mammals and *S. cerevisiae* of Stx5/Sed5p and its well-conserved interactions with cognate ER-Golgi SNAREs, the Sec1/Munc18-like protein Scfd1/Sly1p, and the tethering complexes COG, p115, and GM130, there are several important differences in the function and regulation of Stx5/Sed5p. First, Stx5/Sed5p interacts differently with the COPII coat protein Sec24p in yeast and Sec24C/D in mammalian cells [20,62,71,72], indicating a different mechanism of recruitment to COPII-coated vesicles. Based on sequence alignments (Figure 3 and Figure 4A), one of the two sites of Sed5p binding to Sec24p in yeast seems to be conserved for Stx5 binding to Sec24A/B in mammals, but this remains to be proven. The different binding of Sed5p/Stx5 to Sec24 might relate to differences in COPII function among yeast and mammals. In yeast, COPII mediates anterograde trafficking from the ER directly to the *cis*-Golgi [7], whereas in mammals, COPII vesicles fuse together to form the ERGIC [2,4]. Second, Stx5 is present as two isoforms in mammalian cells, but there is only one short isoform of Sed5p in yeast. The long isoform of Stx5 interacts with microtubules [68], which correlates with the finding that the microtubular cytoskeleton is involved in ER-Golgi trafficking in mammalian cells, but not in yeast [2,4,5]. Moreover, the long isoform of Stx5 is retrieved to the ER, while the short isoform of Stx5 locates more to the Golgi [49,57,58]. These findings suggest that in mammalian cells, the long isoform of Stx5 has a role in early ER and *cis*-Golgi trafficking, whereas the short isoform of Stx5 is involved in later trafficking at the medial- and *trans*-Golgi. These potentially distinct trafficking roles of Stx5 isoforms might contribute to how mammalian cells maintain a complex Golgi structure of intercalated tubular networks of *cis*-, medial, and *trans*-Golgi intercalating into a large network, whereas in yeast, with only a single Sed5p isoform, the Golgi is organized into discrete cisternae that are scattered throughout the cell [2,4,5]. An intriguing possibility that thus emerges from this review is that the different organization of ER-Golgi trafficking between yeast and mammalian cells might be partly attributable to the interactions of a single SNARE protein: Stx5/Sed5p, although there are presumably other reasons for this such as the differential role of microtubules and the larger number of proteins involved in ER-Golgi trafficking in mammals versus yeast. This might not be surprising, given the rate-limiting function of Stx5/Sed5p in ER-Golgi transport [35] and its central role in the organization of the Golgi structure [15,31,32]. Moreover, Stx5 is tightly regulated during cell division [65,73,74,99] and is hijacked by several pathogens to promote their replication within infected cells [102,103,104,109], and dysregulation of Stx5-mediated trafficking is implicated in neurodegenerative diseases [57,105,106]. Thus, Stx5/Sed5p plays a pivotal role in the early secretory pathway, and understanding its function is important for understanding the organization of eukaryotic cells and disease mechanisms.

## Figures and Tables

**Figure 1 cells-08-00780-f001:**
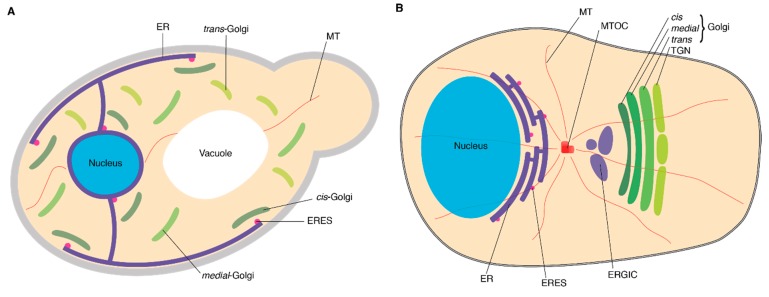
Schematic overview of the early secretory pathway in *Saccharomyces cerevisiae* (**A**) and mammalian cells (**B**). Abbreviations: ER, endoplasmic reticulum; ERES, endoplasmic reticulum exit sites; ERGIC, endoplasmic reticulum-Golgi intermediate compartment; MT, microtubule; MTOC, microtubule organizing center; TGN, *trans*-Golgi network.

**Figure 2 cells-08-00780-f002:**
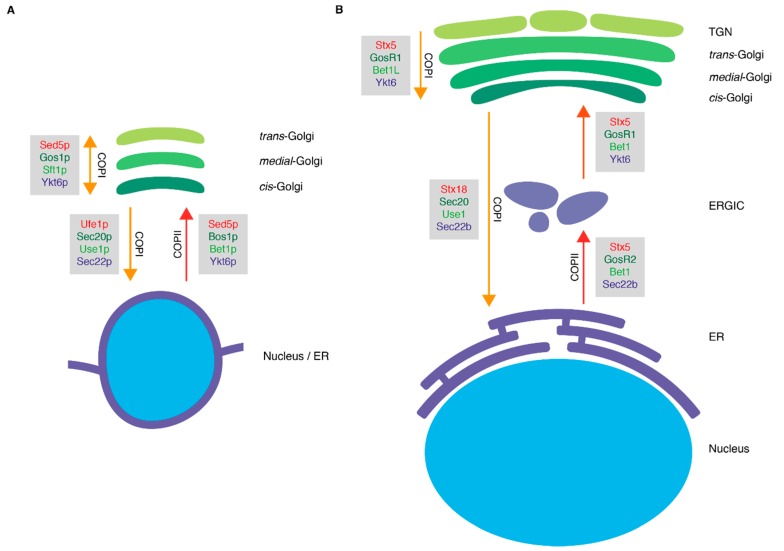
Schematic overview of SNARE complexes of the early secretory pathway in *Saccharomyces cerevisiae* (**A**) and mammalian cells (**B**). The grey boxes indicate the known SNARE complexes and their location along the secretory pathway. Colors of the SNAREs: red, Qa-SNAREs; dark green, Qb-SNAREs; light green, Qc-SNAREs; blue, R-SNAREs. Abbreviations: ER, endoplasmic reticulum; ERGIC, endoplasmic reticulum-Golgi intermediate compartment; TGN, *trans*-Golgi network.

**Figure 3 cells-08-00780-f003:**
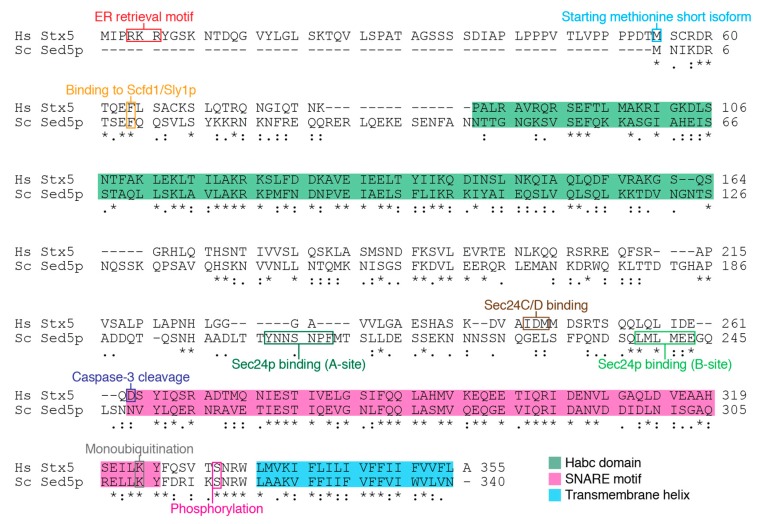
Conserved and distinct interactions of mammalian Stx5 and yeast Sed5p. Alignment of human (Hs) Stx5 and *S. cerevisiae* (Sc) Sed5p. Indicated are: the double arginine ER retrieval motif of the long isoform of Stx5 [49,57]; the alternative starting methionine of the short isoform of Stx5 [49]; the binding site to Scfd1 (mammals) and Sly1p (yeast) [60,61]; Sec24C/D binding site to the IxM motif of Stx5 [20,62]; Sec24p binding sites to the YNNSNPF motif (A-site) and LxxME motif (B-site) of Sed5p [63]; Caspase-3 cleavage site of Stx5 [64]; monoubiquitination site of Stx5 [65]; phosphorylation site of Sed5p [66].

**Figure 4 cells-08-00780-f004:**
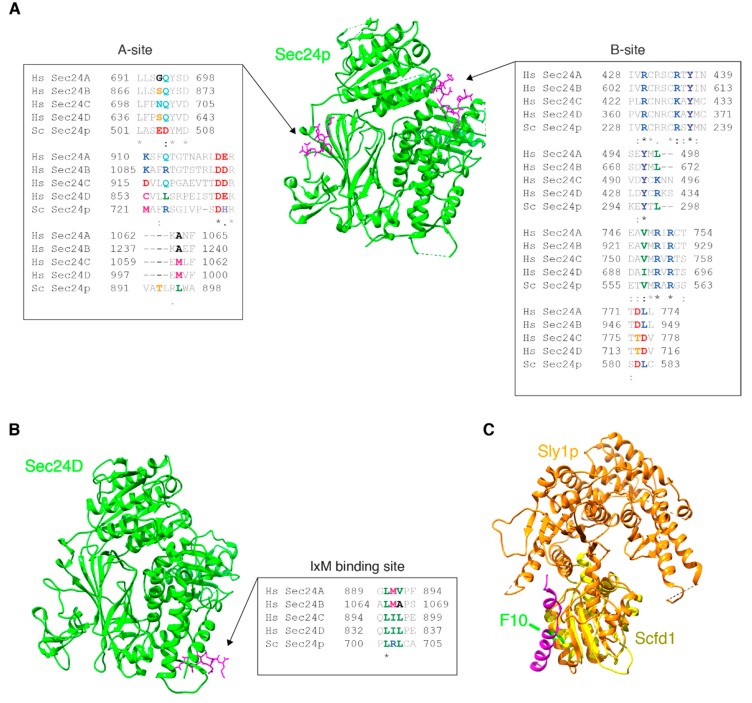
Interactions of Stx5/Sed5p with Sec24 and Scfd1/Sly1p. (**A**) Crystal structure of yeast Sec24p (green) bound to the YNNSNPF motif of Sed5p (A-site) and the LxxME motif of Bet1p (B-site) [63]. Alignments of interacting regions of Sec24p with mammalian Sec24A–D are shown. Substrate-interacting residues are in bold and colored. Note that the A-site is not conserved, whereas the B-site seems conserved in mammalian Sec24A and B (not C and D). (**B**) Crystal structure of mammalian Sec24D (green) with the IxM motif of Stx5 (magenta; residues 241–248) [62]. Note that the IxM binding site is not conserved in mammalian Sec24A–D. (**C**) Conserved interactions between Stx5/Sed5p and Scfd1/Sly1p. Crystal structure of yeast Sly1p (orange) with the N-terminal region of Sed5p (magenta; residues 1–21) [61] aligned with an NMR structure of mammalian Scfd1 (yellow) [60]. Interacting residue F10 of the short isoform of Stx5 and Sed5p is indicated [60,61].

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
