# Peer review of "Stx5-Mediated ER-Golgi Transport in Mammals and Yeast"

_cells, 2019, doi:10.3390/cells8080780_

Round 1

Reviewer 1 Report

I'm not a fan of the comprehensive review that covers absolutely everything - I much prefer a focussed review.  I rather enjoyed this one, with a focus on syntaxin-mediated ER-Golgi trafficking of cargo, highlighting the differences between Stx5 and Sed5p and tying these to the differences between COPI and COPII mediated ER-Golgi transport in mammalian cels and yeast.

Figure 4 should be improved for clarity:  the machine read-out for the sequencs of the A B and IxM binding sites is in parts unreadbale. In particular the yellow and pale grey fonts are almost invisible.

Author Response

I'm not a fan of the comprehensive review that covers absolutely everything - I much prefer a focussed review.  I rather enjoyed this one, with a focus on syntaxin-mediated ER-Golgi trafficking of cargo, highlighting the differences between Stx5 and Sed5p and tying these to the differences between COPI and COPII mediated ER-Golgi transport in mammalian cels and yeast.

Figure 4 should be improved for clarity:  the machine read-out for the sequencs of the A B and IxM binding sites is in parts unreadbale. In particular the yellow and pale grey fonts are almost invisible.

We amended Figure 4 as requested by the reviewer.

Reviewer 2 Report

This review is well written and informative and I belive is likely to be of interest to the field. The accompanying figures are clear and nicely complement the text. I am happy to support publication, although a few points need to be addressed, as detailed below:

In the abstract, introduction and discussion, it is argued that differences in Stx5 interactions in yeast and mammals underlie the differences in secretory pathway organization in the two cell types. This seems like an over-statement to me. There are presumably many other reasons for this e.g. the role of microtubules in mammals versus yeast, the larger number of proteins involved in trafficking in mammals versus yeast and so on. The authors may want to tone down this statement. It may help contribute to the different organization, but it is not likely to be the sole reason for it.

At the bottom of page 1/top of page 2 it is stated that COPI vesicles mediate ERGIC to cis-Golgi transport in mammals. This is not correct as far as I am aware. 

There is an entire section entitled SNAREs in ER to Golgi transport, yet the text in this section also mentions intra-Golgi transport, which is also shown in the diagrams. The title should be changed, or perhaps better would be to split the information, with a separate section on SNAREs in intra-Golgi transport.

Regarding intra-Golgi transport, the text does a poor job of describing how cargo is transported, and what the role of retrograde transport is. There is no mention of cisternal maturation, for example. It is claimed that some SNARE complexes may be involved in anterograde transport while others are in retrograde transport (lines 126-132). The papers cited here are all quite old, and do not fit with the current consensus that COPI vesicles predominantly mediate retrograde transport.  This part of the review needs rewriting to better convey how cargo is transported in the Golgi, and what the role of intra-Golgi COPI vesicle transport is. This is particularly important considering the different SNARE complexes found in the Golgi, and how their perturbation and interactions influence trafficking. Related to this point, it is stated that Stx5 mediates anterograde transport in Drosophila, and reference 30 is cited. This does not appear to be correct- the paper does not show what is claimed in the text.

There is an issue with the definition of non-canonical ER export and secretory trafficking, as used in the review (line 133 and also later-line 267). Pro-collagen (PC) is transported via COPII carriers. These may have a different morphology to the small COPII coated vesicles originally described, but I don’t think this fact justifies the use of non-canonical, especially since there is unconventional secretion of other cargoes, which may result in some confusion. The wording should be changed. I also think there could be a better description of what Stx5 is doing in the transport of PC, VLDL and chylomicron- is it ER exit, or in the fusion of the carriers with the ERGIC or Golgi? 

In the introduction and later in the main text (lines 35 and 382), the text states S. cerevisiae has mini-stacks. This is not technically correct. The cisternae in this yeast are not stacked at all, but exist as discrete cisternae. This needs correcting. The authors may also want to mention that in other yeasts e.g. P. pastoris (budding) and S. pombe, the Golgi is indeed present as mini-stacks that are dispersed in the cytoplasm.

The text on lines 224/5 mentions conservation of the B-site in Sec24A and Sec24B, suggesting that mammalian Sec 24A/B can bind Stx5, similar to what occurs in yeast. Has this been tested? If it were correct, would this change the view of how we think about Stx5 recruitment to COPII vesicles compared to that of Sed5p? It is argued in the text that COPII interaction is different between mammalian Stx5 and yeast Sed5p, but maybe this is not correct if both bind Sec 24A/B?

Ref 71 in the reference list is wrongly described.

Line 258: What is the significance of Sed5p phosphorylation? Is it in response to growth signals? Some additional explanation is required here.

The section on Scdf1/Sly1p would be better with some sort of synthesis, or broader conclusion as to what these proteins do. Is there a common consensus as to their mechanism of action and cellular function?

Line 299: A general review for golgin function should be cited here.

The text on lines 301-304 is inconsistent. It states that COG mediates intra-Golgi transport then also states that the tethering complexes mediate specificity of ER to Golgi transport, without mentioning specificity of intra-Golgi transport. The text on the roles of tethers (and SNAREs) in ER to Golgi versus intra-Golgi transport needs to clarified.

Line 334: the text describing the model for GM130 binding to Stx5 is not quite right. GM130 binding to Stx5 is proposed to prevent its interaction with other SNAREs. Binding of p115 to GM130 then releases Stx5 from GM130, allowing it form SNARE complexes downstream of tethering, and fusion to occur. Hence, in mitosis, stabilizing the GM130-Stx5 interaction would be expected to reduce membrane fusion.

Author Response

This review is well written and informative and I belive is likely to be of interest to the field. The accompanying figures are clear and nicely complement the text. I am happy to support publication, although a few points need to be addressed, as detailed below:

In the abstract, introduction and discussion, it is argued that differences in Stx5 interactions in yeast and mammals underlie the differences in secretory pathway organization in the two cell types. This seems like an over-statement to me. There are presumably many other reasons for this e.g. the role of microtubules in mammals versus yeast, the larger number of proteins involved in trafficking in mammals versus yeast and so on. The authors may want to tone down this statement. It may help contribute to the different organization, but it is not likely to be the sole reason for it.

As requested by the reviewer, we toned down our statement in the Abstract, Introduction and Discussion sections. We now write that the differences in Stx5/Sed5p contributeto the different organization of ER-Golgi trafficking, but that other factors are likely involved in this as well.

At the bottom of page 1/top of page 2 it is stated that COPI vesicles mediate ERGIC to cis-Golgi transport in mammals. This is not correct as far as I am aware. 

There is clear evidence based on microinjection of COPI-targeted antibodies and live cell imaging that COPI mediates anterograde transport, as discussed in [Appenzeller-Herzog, (2006) J. Cell Sci. 119:2173], although we agree it has not been proven directly. We now added a citation to the aforementioned review. 

There is an entire section entitled SNAREs in ER to Golgi transport, yet the text in this section also mentions intra-Golgi transport, which is also shown in the diagrams. The title should be changed, or perhaps better would be to split the information, with a separate section on SNAREs in intra-Golgi transport.

We changed the title of the section to also include intra-Golgi transport, as requested by the reviewer.

Regarding intra-Golgi transport, the text does a poor job of describing how cargo is transported, and what the role of retrograde transport is. There is no mention of cisternal maturation, for example. It is claimed that some SNARE complexes may be involved in anterograde transport while others are in retrograde transport (lines 126-132). The papers cited here are all quite old, and do not fit with the current consensus that COPI vesicles predominantly mediate retrograde transport.  This part of the review needs rewriting to better convey how cargo is transported in the Golgi, and what the role of intra-Golgi COPI vesicle transport is. This is particularly important considering the different SNARE complexes found in the Golgi, and how their perturbation and interactions influence trafficking. Related to this point, it is stated that Stx5 mediates anterograde transport in Drosophila, and reference 30 is cited. This does not appear to be correct- the paper does not show what is claimed in the text.

We have added an explanation of the cisternal maturation model, and mentioned the caveat that the Golgi gradient model does not align with the cisternal maturation model. Regarding the role of Stx5 in anterograde transport in Drosophila, we cited the wrong reference here. This should be [Satoh, et al. (2016) Biol. Open 5:1420]; we corrected this and apologize for the mistake.

There is an issue with the definition of non-canonical ER export and secretory trafficking, as used in the review (line 133 and also later-line 267). Pro-collagen (PC) is transported via COPII carriers. These may have a different morphology to the small COPII coated vesicles originally described, but I don’t think this fact justifies the use of non-canonical, especially since there is unconventional secretion of other cargoes, which may result in some confusion. The wording should be changed. I also think there could be a better description of what Stx5 is doing in the transport of PC, VLDL and chylomicron- is it ER exit, or in the fusion of the carriers with the ERGIC or Golgi? 

As requested by the reviewer, we removed the term non-canonical. We now write that Stx5 is involved in the ER-Golgi transport of specialized cargo molecules. We also clarified the precise roles of Stx5 in the ER-Golgi trafficking of pro-collagens, VLDL and chylomicron.

In the introduction and later in the main text (lines 35 and 382), the text states S. cerevisiae has mini-stacks. This is not technically correct. The cisternae in this yeast are not stacked at all, but exist as discrete cisternae. This needs correcting. The authors may also want to mention that in other yeasts e.g. P. pastoris (budding) and S. pombe, the Golgi is indeed present as mini-stacks that are dispersed in the cytoplasm.

We changed the text as suggested by the reviewer.

The text on lines 224/5 mentions conservation of the B-site in Sec24A and Sec24B, suggesting that mammalian Sec 24A/B can bind Stx5, similar to what occurs in yeast. Has this been tested? If it were correct, would this change the view of how we think about Stx5 recruitment to COPII vesicles compared to that of Sed5p? It is argued in the text that COPII interaction is different between mammalian Stx5 and yeast Sed5p, but maybe this is not correct if both bind Sec 24A/B?

The reviewer is correct that, based on sequence alignments (figure 3, 4A), we predict that one of the two sites of Sed5p binding to Sec24p in yeast seems to be conserved for Stx5 binding to Sec24A/B in mammals. However, this prediction remains to be proven by binding assays. Nevertheless, the second binding site of Sec24p is not conserved and the binding of Stx5 to Sec24C/D in mammals completely differs from Sed5p binding to Sec24p in yeast, which is why we believe it is warranted to claim that the recruitment of Sed5p/Stx5 is very different between mammals and yeast. We now explain this better in the text (Discussion section).

Ref 71 in the reference list is wrongly described.

We corrected this reference to: 

[Lowe, M.; Lane, J.D.; Woodman, P.G.; Allan, V.J. Caspase-mediated cleavage of syntaxin 5 and giantin accompanies inhibition of secretory traffic during apoptosis. J. Cell Sci. 2004, 117, 1139–1150]

Line 258: What is the significance of Sed5p phosphorylation? Is it in response to growth signals? Some additional explanation is required here.

The precise role of Sed5p phosphorylation is not known, but it is speculated that it might play a role in Golgi inheritance during mitosis as it allows for the Golgi to cycle between ordered and dispersed states. We added this explanation to the text.

The section on Scdf1/Sly1p would be better with some sort of synthesis, or broader conclusion as to what these proteins do. Is there a common consensus as to their mechanism of action and cellular function?

As suggested by the reviewer, we added a general conclusion/synthesis of the role of Sec1/Munc18-like proteins and added a citation to a general review on this topic [Carr & Rizo. (2010) Curr. Opin. Cell Biol. 22:488].

Line 299: A general review for golgin function should be cited here.

We added a citation to a relevant review [Cottam & Ungar. (2012) Protoplasma 249:943].

The text on lines 301-304 is inconsistent. It states that COG mediates intra-Golgi transport then also states that the tethering complexes mediate specificity of ER to Golgi transport, without mentioning specificity of intra-Golgi transport. The text on the roles of tethers (and SNAREs) in ER to Golgi versus intra-Golgi transport needs to clarified.

COG has been described to regulate intra-Golgi trafficking, and not ER-Golgi trafficking. We corrected the text and apologize for the mistake.

Line 334: the text describing the model for GM130 binding to Stx5 is not quite right. GM130 binding to Stx5 is proposed to prevent its interaction with other SNAREs. Binding of p115 to GM130 then releases Stx5 from GM130, allowing it form SNARE complexes downstream of tethering, and fusion to occur. Hence, in mitosis, stabilizing the GM130-Stx5 interaction would be expected to reduce membrane fusion.

We corrected the text as requested by the reviewer.